# Nanomaterials for Skin Cancer Photoimmunotherapy

**DOI:** 10.3390/biomedicines11051292

**Published:** 2023-04-27

**Authors:** Carlota M. Relvas, Susana G. Santos, Maria J. Oliveira, Fernão D. Magalhães, Artur M. Pinto

**Affiliations:** 1LEPABE—Laboratory for Process Engineering, Environment, Biotechnology and Energy, Faculdade de Engenharia, Universidade do Porto, 4200-465 Porto, Portugal; 2ALiCE—Associate Laboratory in Chemical Engineering, Faculdade de Engenharia, Universidade do Porto, 4200-465 Porto, Portugal; 3i3S—Instituto de Investigação e Inovação em Saúde, Universidade do Porto, Rua Alfredo Allen, 208, 4200-180 Porto, Portugal; 4INEB—Instituto de Engenharia Biomédica, Universidade do Porto, Rua Alfredo Allen, 208, 4200-180 Porto, Portugal

**Keywords:** immunotherapy, photothermal therapy, photodynamic therapy, melanoma, basal cell carcinoma, squamous cell carcinoma

## Abstract

Skin cancer is one of the most common types of cancer, and its incidence continues to increase. It is divided into two main categories, melanoma and non-melanoma. Treatments include surgery, radiation therapy, and chemotherapy. The relatively high mortality in melanoma and the existing recurrence rates, both for melanoma and non-melanoma, create the need for studying and developing new approaches for skin cancer management. Recent studies have focused on immunotherapy, photodynamic therapy, photothermal therapy, and photoimmunotherapy. Photoimmunotherapy has gained much attention due to its excellent potential outcomes. It combines the advantages of photodynamic and/or photothermal therapy with a systemic immune response, making it ideal for metastatic cancer. This review critically discusses different new nanomaterials’ properties and mechanisms of action for skin cancer photoimmunotherapy and the main results obtained in the field.

## 1. Introduction

Skin cancer is the out-of-control growth of abnormal skin cells [1,2]. It is one of the most frequent types of cancer, being responsible for over 19 million new cases of cancer and almost 10 million deaths worldwide in 2020 [3]. Concerning its geographic distribution, North America is estimated to represent the majority of the cases (45.4%), followed by Europe (28.4%), Central and South America (9.5%), Australia, and New Zealand (5.8%). In Europe, with 150,627 new cases of melanoma and over 356,180 non-melanoma skin cancers. In 2022, melanoma caused 26,360 deaths, while for non-melanoma skin cancers there were 12,679 deaths, in Europe [4]. The continuous incidence of skin cancer is due to several factors, being the main reason for ultraviolet light exposure [5,6,7], but also due to environmental and hereditary risk factors [8,9]. At the pathophysiological level, it is divided into two groups: melanoma and non-melanoma tumors. While melanoma derives from melanocytes, non-melanoma tumors derive from cells from the epiderm. The most common skin tumors are basal cell carcinomas, the most frequent of them arising from interfollicular epidermis exhibiting mutations on TP53 gene; squamous cell carcinoma with origin from squamous keratinocytes in the epidermis of the skin or mucous membranes; and melanoma with origin from melanocytes present in the skin, which incidence every year is notably increasing. Particularly in the United States (U.S.), the economic burden of this disease was USD 8.9 billion in recent years (2016–2018). About USD 6.5 billion were related to non-melanoma tumors, whereas approximately USD 2.5 billion were associated with melanoma tumors. The latter is expected to triple by 2030 as a result of the overall rising of healthcare costs [10]. The type of treatment chosen depends on tumor type, location, and progression degree. There is a wide variety of treatment options, starting with surgical treatment, such as excision surgery, Mohs surgery and curettage, and cryotherapy, radiotherapy, chemotherapy, and photodynamic therapy [11,12,13,14,15]. Although there are already several treatment options available, its high incidence makes skin cancer an important health problem and relevant to the study and development of new approaches for skin cancer management [1]. New therapies can include cancer immunotherapy, cancer photodynamic and/or photothermal therapy, and cancer photoimmunotherapy. More recently, new approaches, such as the use of metallic, polymeric, and lipid-based nanotheranostics, emerged in this field to facilitate the treatment and diagnosis of this disease. These platforms may serve as contrast agents with therapeutic properties (e.g., reactive oxygen species production). Moreover, the versatility of their formulations allows the incorporation of other materials (e.g., gold) and/or drugs (e.g., parvifloron D), which act synergistically to impair the development of solid and superficial tumors, namely at early stages [16,17]. All the above mentioned new therapies and nanomaterials mechanisms of action are discussed in more detail in the next sections.

### 1.1. Immunotherapy

Cancer immunotherapy has been contributing to improved survival and quality of life for cancer patients. It consists in triggering the immune system to control and fight cancer, overcoming the mechanisms that cancer cells develop to escape immune surveillance and avoid detection and elimination [18,19]. Immunotherapy started more than 100 years ago, in New York, with Dr. Coley, who treated sarcoma patients with Coley’s toxin, a vaccine with a mixture of two bacterial toxins [20]. Currently, there are two types of immunotherapies, active and passive immunotherapy. Active immunotherapy stimulates the patient’s immune system, whereas passive immunotherapy can be with the administration of, for example, cytokines, vaccines, and antibodies [21,22,23].

The immune system plays a critical role in recognizing, eliminating, and controlling tumor progression. However, cancer cells develop mechanisms to avoid it, namely: (i) downmodulation of components of antigen processing and presentation machinery; (ii) an environment that promotes suppressor immune cells, such as regulatory T cells (Treg), an immunosuppressive subset of CD4+ T-cell family, myeloid-derived suppressor cells (MDSCs), and tumor-associated macrophages, which are anti-inflammatory macrophages (M2- like); (iii) production of soluble factors associated with immunosuppression, such as TGF-β and IL-10; (iv) and upregulation of ligands for coinhibitory receptors that downmodulate programmed death ligand-1 (PD-L1) [24,25,26,27,28,29].

Dendritic cells (DCs) induce the differentiation of T cells to their antigen-specific effector T cells. CD4+ T cells are responsible for inducing DC maturation and for CD8+ T-cell priming. The primed cells are activated to form cytotoxic T lymphocytes, and these are responsible for releasing INF-γ and TNF-α, which will induce cytotoxicity in cancer cells. INF-γ is produced by both CD4+ and CD8+ T cells and stimulates the antitumor pro-inflammatory macrophages (M1). These tumor suppressor cells, such as cytotoxic T lymphocytes, also upregulate the release of pro-inflammatory cytokines, namely, IL-2, IL-6, IL-12, INF-γ, and TNF-α [24,30,31,32]. The higher calreticulin (CRT) exposure and release of high mobility group box 1 (HMGB-1) also act as an “eat-me” signal to induce tumor cell apoptosis [33,34,35,36].

### 1.2. Phototherapy for Cancer Treatment

Phototherapy (PT) started 4000 years ago in ancient Egypt to treat Vitiligo, when a plant extract was boiled and then combined with sun exposure. Modern phototherapy, on the other hand, started only in the 70s, using artificial light sources [37,38,39]. Phototherapy is mainly divided into two categories, photodynamic therapy (PDT) and photothermal therapy (PTT). In PDT, a photosensitizer agent is irradiated by light to generate reactive oxygen species (ROS). These are highly toxic, causing cell death. PTT is based on local temperature increase, usually triggered by laser radiation. Usually, NIR lasers (650–1350 nm) are used in PTT due to their efficiency in penetrating tumors [40,41,42,43]. PTT can be divided into two categories. In the first, called mild hyperthermia, temperature increases up to 43–50 °C, leading to enhanced membrane permeability, cellular uptake, metabolic signaling disruption, and dysfunctional membrane transport. The capability of tumor cells to recover from such damages is very low. The second one is photothermal ablation (>50 °C), which destroys the cellular membrane, leading to necrotic cell death [44]. PDT has been FDA-approved for almost 40 years [40,45]. Hematopotphyrin derivative (HPD) was the first PS receiving FDA approval, nowadays Foscan^®^, Levulan^®^, Radachlorin^®^, Metvix^®^, and Photofrin^®^ are FDA-approved PS [39,43].

### 1.3. Photoimmunotherapy for Cancer Treatement

PT triggers immunogenic cell death (ICD) that will release tumor-specific antigens (TSAs) and damage-associated molecular patterns (DAMPs), namely CRT, HMGB-1, and ATP. This phenomenon increases the immunogenicity of the tumor microenvironment once DAMPs induce the maturation of DCs, and pro-inflammatory cytokines, such as IL-2, IL-6, IL-12, INF-γ and TNF-α, were also reported to increase. The immunostimulatory effect of PT boosts anti-tumor immunity when compared to immunotherapy alone. Although immunotherapy by itself can be effective in triggering the immune response at tumor site, it is inefficient to eradicate primary tumors [46,47,48,49,50].

When combining phototherapy with immunotherapy, in photoimmunotherapy (PIT), a synergy is reported to occur between them. PT directly kills the tumor cells and triggers a systemic immune response, and when in combination with immunotherapy, immunological memory is formed. Photoimmunotherapy has the advantages of phototherapy and the ability to trigger an immune response, making it ideal for treating metastatic cancer. Thereby, PIT eradicates primary tumors and, through simultaneously stimulating immune memory, it has the potential to prevent tumor recurrence and metastasis [42,46,51,52].

## 2. Nanomaterials

### 2.1. Structure and Properties

In recent years, nanomaterials have gained much interest in the biomedical field, namely in drug delivery, tissue engineering, diagnosis, and theragnostics, amongst others. Nanomaterials can be divided into different categories according to their properties (e.g., size, shape, physicochemical properties, etc.). Regarding nanomaterials used for skin cancer photoimmunotherapy, the focus of this review, the main categories found are metallic, polymeric, lipid-based and 2D nanomaterials [53], as illustrated in Figure 1.

Metallic nanoparticles are nanosized metals which present a metal core covered in a metal shell. Their size can range from 1 up to 100 nm. Surface modification strategies are employed on metallic nanoparticles to prevent their agglomeration in physiological conditions. Nanoparticles with a zeta potential greater than 30 mV and lower than −30 mV are more water stable. Additionally, particle shape is crucial for the biological response because it affects the ability of particles to cross biological barriers. The most common shapes are rods, spheres, cylinders, and cubes. Spherical shapes are reported to have less toxicity than star-shaped counterparts on human skin fibroblasts. Branched-shaped nanoparticles were reported to have high toxicity due to their ability to cling to cells for longer periods of time, causing more damage [54,55,56,57,58,59].

Polymeric nanoparticles can present a nanocapsule or a nanosphere morphology. Nanocapsules have an oily core surrounded by a thin polymeric envelope. Nanospheres are made of a solid polymeric network. A PS or a drug can be dissolved in the core or adsorbed to the polymeric wall in polymeric nanoparticles. The drug release should be controlled temporal and spatially to reduce side effects and prolong therapeutic effects as desired. Drug diffusion can be affected by the presence of water because the pores of the polymeric matrix become enlarged, allowing drug diffusion. Additionally, the thickness of the polymeric envelope and the permeability of the drug impact its release through the polymeric matrix. The degradation rate of the polymer is also very important [60,61,62,63]. Polymeric nanoparticles size usually ranges from 100 to 300 nm, but dimensions under 100 nm have also been reported. The polymers used can have a crystalline or amorphous structure, which will impact the physicochemical properties of the nanoparticles, including their drug release performance [64,65].

Lipid-based nanoparticles can be liposomes that have a core-shell nanostructure making them appropriate for loading hydrophobic and hydrophilic molecules. Hydrophobic molecules are encapsulated in the lipophilic bilayers of the shell, and hydrophilic molecules are entrapped in their core. They can also be solid lipid nanoparticles (SLN), composed of a solid lipid core stabilized in aqueous media by surfactants. SLN usually present improved stability, protection, and controlled release. However, the drug loading capacity is limited by their polymorphism. The nanostructured lipid carriers (NLC) are composed of a mixture of liquid and solid lipids, with a liquid lipid range between 10 and 30%. NLCs have improved drug retention and enhanced drug loading capacity and are divided into: (i) imperfect type with a highly disordered matrix leading to high drug loading capacity but low encapsulation efficiency; (ii) amorphous type, that contains a structureless solid matrix to avoid crystallization-induced drug leakage; (iii) multiple type with oil nanocompartments distributed in the solid matrix that acts as a barrier preventing drug leakage and controlling drug release. The drug solubility in an oil nanocompartment is higher than in a solid matrix enabling higher drug loading [66,67,68,69,70,71]. Their size range between 10 and 1000 nm, but there are also lipid-based nanoparticles between 10 and 150 nm, the ideal ones for drug delivery since they can transport hydrophobic and hydrophilic molecules, can be surface modified and offer controlled drug release [66].

Two-dimensional nanomaterials have a sheet-like structure and lateral dimensions larger than 100 nm but a thickness usually below 5 nm. They have a large surface area and anisotropic physical and chemical properties. Two-dimensional nanomaterials can be, for example, carbon-based materials (e.g., graphene, graphene oxide), clay materials, transition metal oxides or black phosphorus. These materials can be layered: the layers stack together due to the weak Van der Waals interaction, and inside them, strong chemical bonds connect the atoms. The arrangement of the atoms determines the physicochemical properties. Two approaches may be applied to prepare 2D layered nanomaterials: (i) the top-down approach uses driving forces to break the van der Waals interactions between layers and achieve exfoliation into 2D primary sheets; (ii) the bottom-up one relies on assembly of 2D nanosheets following an appropriate chemical synthesis route [72,73,74,75,76,77].

### 2.2. Surface Modification and Encapsulation Strategies

Nanomaterials to be used for biomedical applications should have good stability in physiological conditions and biocompatibility. Many nanoparticles agglomerate in aqueous solutions due to being hydrophobic or creating strong interparticle interactions, for instance. Covalent and non-covalent surface functionalization can be performed to enhance nanoparticles properties. However, such modifications should not affect nanomaterials photoabsorption properties if skin cancer photoimmunotherapy applications are desired. Figure 2 shows examples of such approaches.

Functionalization with PEG usually increases the nanoparticles biocompatibility and blood circulation time. It also decreases the interaction with the targeted cells [78,79,80,81,82]. It is relevant to mention that functionalization often changes nanomaterials physicochemical properties, such as conformation, electrostatic binding, and hydrophobicity, which might impact not only its biointeractions but also its photo/immunotherapy effectiveness. There are already approved PEGylated drugs by FDA, such as Adagen^®^ and Oncaspar^®^ [83]. Other synthetic polymers used for surface coating include polyethyleneimine (PEI), polyvinyl alcohol (PVA), and polyvinylpyrrolidone (PVP). Some other surface modifications strategies include the use of polysaccharides, namely, chitosan or hyaluronic acid. These act as steric protection against protein adsorption and macrophage uptake. They are also biocompatible and biodegradable. When coating metallic nanoparticles, they can also help prevent oxidation and release of metal ions [78,79,80,81,82].

Antibodies possess high target specificity and affinity for certain molecules at the cell surface. They can be used, for example, to direct particles for selective drug delivery or phototherapy. Furthermore, the potential high recognition capabilities of antibody-modified particles might be able to reduce immunological side effects [78,79,80,82,84]. Aptamers are synthetically prepared short oligonucleotides. They can interact with cellular targets (e.g., nucleic acids, transmembrane proteins, sugars) with high affinity and selectivity. They present advantages over antibodies, such as having a smaller size, a higher ratio of target accumulation, and higher in vivo stability. However, their half-time circulation is shorter [79,80,82,85]. Short peptides have also been used for surface functionalization because they offer several advantages, such as increased stability, reduced immunogenicity, and versatile methods for conjugation with nanoparticles. Small molecules can recognize some markers or receptors on the surface of target cells, increasing nanoparticle uptake [79,80,82].

Lipids are another surface modification strategy. Lipid-coated nanoparticles present several advantages, such as reduced cytotoxicity and better target specificity for drug delivery. Natural phospholipids are often used to coat polymeric nanoparticles. Their amphiphilic nature allows the formation of membrane-mimetic structures on nanoparticles [79,80,82,86].

Encapsulation is the method of enclosing molecules inside a shell or carrier, which can be polymers or lipids. It can be used to protect the molecules from degradation, preserve their properties as well as control their release in the body. Encapsulation can help to deliver the molecules effectively, at a controlled rate, helping to reduce the potential side effects. There are several encapsulation techniques, and the appropriate choice depends on the specific molecule, such as its size, solubility, surface charge, and the desired therapeutic effect. Examples of encapsulation techniques to produce polymer-based nanoparticles are nanoprecipitation or emulsion evaporation; for liposomes, production techniques such as hydration of a thin film or solvent injection can be used [87,88,89]. Nanoprecipitation is a simple technique where non-water-soluble polymer and drug(s) are dissolved in the same organic solvent. The resultant dispersion is added to an aqueous solution that contains a surfactant under stirring, and the solvent diffuses into the aqueous phase, causing the polymer to precipitate and form the nanoparticles [88,90]. The emulsion evaporation method consists in dissolving the polymer and drug in a water-immiscible organic solvent that is later added to an aqueous phase containing a suitable stabilizer. Then, the organic solvent is usually evaporated under stirring to obtain the nanoparticles. High-shear mixing can be used to decrease particle size to the nanometric scale and to narrow size distribution [88,89]. The hydration of a thin film technique involves dissolving the lipids in an organic solvent; the solvent is then evaporated to form a thin lipid film. Finally, hydration with water or a buffer is performed to enable the formation of a liposome dispersion. Sonication is usually performed to increase uniformity and reduce particle size. The solvent injection technique consists of dissolving phospholipids in an organic solvent (e.g., ethanol, ether) that is then injected into an aqueous buffer solution. Lipids self-assembly forming liposomes following ethanol dilution or ether evaporation. This method presents the advantage of allowing to obtain of small nanoscale liposomes (usually below 100 nm) with a narrow size distribution, without extrusion or sonication [88].

## 3. Skin Cancer Photoimmunotherapy Studies

Skin cancer is one of the most common cancers. Although there are already several treatment options available, there is still an urgent need to reduce mortality, recurrence rates, and side effects [3,8,91]. Photoimmunotherapy has gained much interest in recent years. It combines the advantages of phototherapy with enhancement of the immune response, resulting in a more effective cancer treatment approach. Figure 3 shows the mechanisms of action of photoimmunotherapy. Table 1 summarizes the state-of-the-art literature regarding nanomaterials used for skin cancer photoimmunotherapy, including their composition, size, and biological effects.

Zhou et al. [92] produced aluminum (Al) hydroxide nanoparticles with spherical shape, loading the photosensitizer chlorin e6 (Ce6). Aluminum hydroxide was added to bovine serum albumin (BSA) to produce a novel nanosystem (Al-BSA-Ce6 NPs) by biomineralization. Al-BSA.Ce6 NPs had uniform size (25.3 ± 2.1 nm) and were tested at various concentrations, ranging from 0.01 to 0.5 μg mL^−1^ being incubated with B16F10 cells for 3 h. After incubation, the cells were irradiated with a NIR laser at 660 nm, irradiance of 0.8 W cm^−2^ for 5 min and incubated for another hour. Results showed that Al-BSA-Ce6 NPs at a concentration of 0.1 μg mL^−1^ could kill approximately 95% of the B16F10 cells through 3-(4,5-dimethylthiazol-2-yl)-2,5-diphenyl-2H-tetrazolium bromide (MTT) assay. CD80 expression was higher when cells were treated with Al-BSA-Ce6 NPs under laser irradiation, and the levels of TNF-α, IL-6 and IL-12p70 were also higher. These results evidenced that Al-BSA-Ce6 NPs could stimulate the immune response and boost the efficacy of photodynamic/photothermal therapy in vitro. Regarding in vivo tests, the nanoparticles were then intravenously injected into B16F10 tumor-bearing mice at a dose of 5.0 mg kg^−1^ and afterwards irradiated using the same conditions as for in vitro tests. At day 15, the tumors had shrunk almost completely (tumor volume ≈ 0 mm^3^), and ≈63% of the mice survived over 100 days. Results also showed that Al-BSA-Ce6 NPs could successfully induce a strong tumor immune response due to TNF-α and INF-γ production increase.

Gold nanoparticles are quite promising for skin cancer photoimmunotherapy due to their excellent properties, such as good biocompatibility, low toxicity, good light-to-heat conversion, good photostability, and tunable surface modifications [98]. Tian et al. [96] used hyaluronic acid (HA) to modify a gold nanorod (AuNR) surface-modified with matrix metalloproteinase-2 (MMP2)-responsive M2pep fusion peptides (M-M2pep), resulting in HA-AuNR/M-M2pep, with a hydrodynamic size of 64.6 nm. B16F10 melanoma cells were incubated with HA-AuNR/M-M2pep at a concentration of 20 μg mL^−1^ for 24 h. The cells were irradiated with a NIR laser (808 nm, 1.5 W cm^−2^) for 2 min after 6 h of incubation. The treatment led to a tumor apoptosis rate of 35.1 ± 1.8 %, detected with Annecin V-FITC, and an increase in CRT-positive cells and of the HMGB1 release. Regarding in vivo tests, C57BL/6 mice were injected with the same line of melanoma cells and treated with AuNR and MMP-2 at concentrations of 10 and 12 mg kg^−1^, respectively. After 6 h of incubation, the cells were irradiated with the same NIR laser for 2 min. The results showed tumor suppression with the lowest tumor weight and survival rate of 67% over 45 days, compared to only 17% without NIR irradiation. It was also reported an increase in DCs maturation, being CD8+ T cells frequency 3.7-fold higher than other groups and the secretion of INF-γ and of TNF-α 6.3- and 5.6-fold higher, respectively. Zhou et al. [97] developed BSA-bioinspired gold nanorods (GNRs) decorated with polyethylene glycol (PEG) and loaded with imiquimod (R837). The resulting nanocomplex mPEG-GNRs@BSA/R837 has an average diameter of 122.1 ± 11.6 nm and a zeta potential of −12.4 ± 0.1 mV. B16F10 cells were treated with GNRs@BSA/R837 at a gold concentration of 11.5 μg mL^−1^ and irradiated with laser (1064 nm, 1.0 W cm^−2^) for 10 min followed by 24 h incubation. The cell viability decreased to ≈27% in the MTS assay, and a higher release of HSP70/β-Actin was observed. Regarding in vivo tests, mice were inoculated with B16F10 cells, treated with GNRs@BSA/R837 at a gold concentration of 300 μg mL^−1^, and after irradiated with the same settings as used in the in vitro studies. This treatment caused an enhanced infiltration of CD8+ T cells and release of IL-6, IL-12, and TNF-α, leading to the highest mice survival rate (Figure 4A). It has also been hypothesized that the combination of PTT with immunotherapy could cause a higher antitumor effect. Considering this, Zhang et al. [98] used B16F10 melanoma cells to generate AuNPs and then shed nanoparticle trapped vesicles to extracellular environment with retained tumor antigens (AuNP@B16F10). The AuNP@B16F10 were further introduced into dendritic cells (DCs) to produce DC-derived AuNPs (AuNP@DC_B16F10_) with a size around 30 nm. Afterwards, AuNP@B16F10 were introduced into dendritic cells (AuNP@DC_B16F10_), presenting a size around 40 nm. Regarding in vivo tests, AuNP@DC_B16F10_ were injected in murine melanoma models at a concentration of 1.35 mg kg^−1^, and after 24 h of incubation, the mice were irradiated with a laser (808 nm, 2.0 W cm^−2^) for 1 min. The treatment was performed 3 times at 3 days of interval. The authors reported distant tumor inhibition, DCs maturation, high infiltration of CD3+ and CD8+ T cells, increased secretion of INF-γ, and release of TNF-α and IL-6. Notably, on the nineteenth day, 50% of the mice were tumor free, and the treatment showed a long-term effect on tumor inhibition and recurrence. Liu et al. [99] loaded gold nanocages (AuNCs) with simvastatin (SV), and the resulting nanoparticles, AuNCs/SV had a size of 50 nm and were tested against B16-F10 melanoma cells. The cells were treated with AuNCs/SV, with a gold concentration of 60 μg mL^−1^ and irradiated with an 808 nm laser with an irradiance of 1.5 W cm^−2^ for 2 min. The treatment led to a higher mature DCs frequency and a 3.5-fold higher CRT expression in vitro. Regarding in vivo tests, C57 mice were inoculated with B16-F10 cells and treated with AuNCs/SV and irradiated with the same settings used in the in vitro tests. The results were consistent with the in vitro results, with improved DC maturation and CD4+ and CD8+ T-cell proliferation being observed. These results emphasize that laser irradiation improves the PTT effect by enhancing the immune response.

Silicon dioxide (SiO_2_) nanoparticles present good biocompatibility, porous structure, very high surface activity and adsorption properties, which makes them an interesting biomaterial to be explored for cancer photoimmunotherapy. Lin et al. [108] loaded the outside surface of SiO_2_ nanoparticles with copper sulfide (CuS) poly((2-di-methylamino)ethyl methacrylate) (PDMAEMA) polycation, producing a gene co-delivery nanoparticle (CSP). CSP was then integrated with the plasmid encoding IL-12 gene(CSP@IL-12). CSP@IL-12 presented an average size of 157 nm and a zeta potential of 31 mV. B16F10 melanoma cells were incubated for 5 h with CSP@IL-12 at a concentration of 35.4 μg mL^−1^ and then irradiated with a laser (1064 nm, 0.65 W cm^−2^) for 5 min followed by 24 h incubation. This treatment caused 89% of cell apoptosis detected with Annexin V-FITC, increased CRT expression, and a DCs maturation of 66%. Regarding in vivo tests, C57BL/6 mice were inoculated with the same melanoma cells and injected with 172.4 μg CSP@IL-12 per mouse. After 24 h incubation, the mice were irradiated with the same settings as used in the in vitro studies for 5 min. The treatment led to prolonged survival, a DC maturation level of approximately 48%, 3-fold higher than the PBS control group, and a higher proportion of CD4+ and CD8+ T-cell infiltration (Figure 4B). The combinatory treatment evidenced a significant increase in the antitumor effects. Li et al. [109] formed UCNP@mSiO_2_ NPs (β-NaYF4:Yb,Er@NaYF4 Core-Shell NPs coated with a silica shell) that were co-loaded with Ce6 and buthionine sulfoximine (BSO) to induce ferroptosis through glutathione depletion, producing UCNP@mSiO_2_@liposome-Ce6-BSO (UCB). UCB had an average size of approximately 50 nm. UCB nanoparticles were used to treat B16/F10 cells at a concentration of 100 μg mL^−1^. The cells were irradiated with a laser (980 nm, 0.7 W cm^−2^) for 10 min. The combined treatment led to the highest apoptosis rate (≈39%), evaluated through Western Blot, and the highest secretion levels of TNF-α, IL-6, and INF-γ. Regarding in vivo tests, C57BL/6 mice were inoculated with the same melanoma cells and injected with UCB at 0.8 mg per mouse. After 4 h, mice were irradiated with the same setting as used in the in vitro studies, but for 20 min now. The authors reported a decreased tumor weight, and higher levels of IL-12p40 and INF-γ. The combinatory treatment boosted the ferroptotic and apoptotic cell death, demonstrating an increase in anti-melanoma efficacy and immune response.

Chen et al. [95] produced a core-shell microneedle (CSMN) array. The photosensitizer indocyanine green (ICG) was encapsulated into chitosan nanoparticles (ICG-NPs), followed by concentrating on the tip shell of microneedles. 1-Methyl-tryptophan (1-MT) was loaded into the cross-linked PVP and poly(vinyl alcohol) (PVA) gel as the microneedle core. The ICG-NPs had a hydrodynamic size of 220 nm. Their zeta potential was 47.2 ± 0.3 mV due to the presence of chitosan’s amine groups. ICG-NPs were tested with concentrations between 10 and 30 μg mL^−1^ against B16 cells and incubated for 2 h, and then irradiated with a laser (808 nm, 0.35 W cm^−2^) for 5 min with additional 24 h incubation. After treatment with a concentration of 30 μg mL^−1^, cancer cell viability was nearly 0% using cell counting kit-8 (CCK-8) assay, the intracellular ROS levels were correlated with the concentration of the nanoparticles; the higher the concentration, the higher the ROS levels release. Additionally, DCs maturation was 3.9-fold higher when treated with ICG-NPs under laser irradiation. These results support that ICG-NPs increased the immunogenicity of dying B16 cells. Despite this, the upregulated indoleamine 2,3-dioxygenase (IDO) expression after PTT aggravates the immunosuppression. Regarding in vivo tests, B16 tumor-bearing C57/BL6 mice were treated with ICG-NPs, followed by laser irradiation during 5 min (808 nm, 0.35 W cm^−2^) after 30 min post-administration, 3 cycles at interval of 2 days were performed. Results showed almost a complete tumor eradication (tumor volume ≈ 0 mm^3^), and 80% of the mice survived without a recurrence for more than 120 days, while the mice in control groups died within 20–40 days. The frequency of CD8+ T cells in distant tumors increased to ≈61%. ICG-NPs induced the maturation of DCs, reaching a level of approximately 55%, much higher than in other groups, followed by an increase in the expression of TNF-α, IL-12p70 and IL-6. These nanoparticles could enhance a durable antitumor response through synergistic immunotherapy.

Chen et al. [102] developed self-assembled nanomicelles dissolving a micro-needle (DMN) patch that was designed for intralesional delivery of immunogenic cell death-inducer (IR780) and chloroquine (CQ) coencapsulated micelles (C/I-Mil). C/I-Mil presented an average diameter between 80 and 90 nm. B16 cells treated with C/I-Mil at 4 μg mL^−1^ under NIR laser irradiation (808 nm, 1.0 W cm^−2^, 5 min). After 24 h incubation, the treatment destroyed cellular membrane integrity, affecting cell-to-cell interaction and migration. This treatment also enhanced the immunogenicity of dying tumor cells. The phagocytic index was evaluated after 16 h incubation, and the results showed it was 3.0-fold higher for C/I-Mil with irradiation, than in control group. All these results proved that this treatment could destroy the cytomembrane integrity and enhance engulfment and cell death. Regarding in vivo tests, B16 tumor-bearing mice, treated with micelles at 4 μg mL^−1^ combined with CQ (20 μg/patch) and NIR irradiation (808 nm, 1.0 W cm^−2^, 5 min) after 45 min incubation, revealed an excellent tumor eradication (tumor volume ≈ 0 mm^3^) with barely any relapse during treatment, and distant tumor volume 3.4-fold smaller. In total, 50% of the mice under this treatment could survive for at least 40 days. The micelles combined with chloroquine were less invasive and could increase the number of immune cells at tumor site. These results highlight the promising antitumor response of this treatment in combination with photothermal therapy. Le et al. [103] used a nanoadjuvant, which comprised imiquimod encapsulated in amphiphilic peptide-based micelles (IQPM), with an average size of 72.0 ± 18.0 nm. Regarding in vivo tests, IQPM was injected at a concentration of 5 mg kg^−1^ in B16F10 tumor-bearing mice. After 24 h of incubation, irradiation was performed with a NIR laser (808 nm, 1.5 W cm^−1^) for 5 min. The treatment led to primary tumor suppression (tumor volume ≈ 0 mm^3^) and an increase in CD8+ and CD4+ T cells, 9.3- and 10.3-fold higher compared to control groups (Figure 5A), followed by a reduction of metastasis. The results demonstrated that the combination of the micelles with NIR irradiation leads to an antitumor immune response, but the underlying molecular mechanisms require further elucidation.

Bian et al. [100] fabricated carrier-free nanoassemblies (NAs) using a Ce6 and α-Linolenic acid (L-Ce6 NAs), which were integrated into a fast-dissolving microneedles (MNs) patch to obtain (L-Ce6 MNs) via a facile fabrication method. L-Ce6 MNs presented sizes with hydrodynamic diameters around 86 nm. B16F10 melanoma cells were treated with L-Ce6 NAs, at a concentration of Ce6 of 400 μM, and incubated for 6 h, followed by laser irradiation (660 nm, 200 mW cm^−2^) for 5 min. L-Ce6 NAs significantly enhanced the expression of CRT (20 h incubation after irradiation), ATP secretion (6 h incubation after irradiation), HMGB1 release (4 h incubation after irradiation), and also the generation of intracellular ROS expression on the surface of B16F10 cells upon laser irradiation. Even if the incubation times post-irradiation were diverse according to the assay, the results were consistent. Regarding in vivo tests, the antitumor effects of L-Ce6 MNs ([Ce6] = 0.12 mg kg^−1^) were tested by administrating them to B16F10 tumor-bearing mice, followed by laser irradiation (660 nm, 200 mW cm^−2^) 4 h post-administration for 4 min. In contrast with untreated or unirradiated groups, mice treated with L-Ce6 MNs followed by laser irradiation showed an enhanced DC maturation. Results also showed an increased proportion of tumor-infiltrating lymphocytes (CD3+) and of CD8+ and CD4+ T cells for L-Ce6 MNs together with laser irradiation. The results showed effective immune responses induced by this treatment.

Li et al. [107] modified polydopamine (PDA) with positively charged polyethyleneimine (PEI) and stabilized by PEGylation to formulate PDA-PEG/PEI (PPP), which endows the complex with CpG oligodeoxynucleotides loading capacity. HA was further coated on PPP by electrostatic interaction to form a core-shell nanoplatform (PPP/CpG/HA). The resulting nanoparticles presented a size of around 140 nm and were used to treat B16F10 tumor cells at a concentration of 200 μg mL^−1^. A total of 12 h after incubation, fresh medium was added to the cells followed by NIR laser irradiation (808 nm, 2.0 W cm^−2^) for 5 min. After 24 h, the expression of CD80 and CD86, both T-cell receptors co-stimulators, was higher when treated with PPP/CpG/HA under laser irradiation, indicating that this treatment could prompt DCs maturation and synergism between PTT and oligodeoxynucleotides (CpG). Regarding in vivo tests, B16F10 tumor-bearing mice were injected with 0.75 mg per kg PPP/CpG/HA followed by laser irradiation (808 nm, 1.5 W cm^−2^) 12 h after administration for 5 min. The results showed that photoablation inhibits primary tumors and leads to an increase in CD8+ and CD4+ T cells at the tumor site, which can also enhance the presence of mature DCs in the tumor (Figure 5B). The synergism between PTT and the immune adjuvant nanoparticles could combat tumor recurrence, lead to longer survival rates, suppress distant tumors and activate potent immune responses.

Song et al. [105] used ionizable block copolymer and acid-liable phenylboronate ester (PBE) dynamic covalent bonds for tumor-specific delivery of the ferroptosis inducer, a glutathione peroxidase 4 inhibitor RSL-3. The resulting nanoparticles (BNP@R) were used to treat B16-F10 melanoma cells at a concentration of a glutathione peroxidase 4 inhibitor RSL-3 and INF-γ of 0.5 μg mL^−1^ and 100 ng mL^−1^, respectively. After that, laser irradiation (671 nm, 100 mW cm^−2^) was performed for 1 min, leading to increased DC maturation and a 5.0-fold higher CRT expression compared to other groups. Regarding in vivo tests, C57BL/5 mice were inoculated with the same cells and treated with the same concentrations. After 6 h of incubation, mice were irradiated with a laser (671 nm, 150 mW cm^−2^) for 2 min. Results revealed a higher secretion of INF-γ, prolonged survival and inhibition of the tumor growth at an early stage (2-fold lower tumor volume). Liu et al. [104] integrated a PEGylated IDO inhibitor and a photosensitizer ICG into a core-shell nanostructure cleaved with MMP-2 (mPEG-Pep-IDOi/ICG nanoparticles), which enhanced the permeability and retention effect. mPEG-Pep-IDOi/ICG were then tested in B16-F10 cells, incubated with ICG at a concentration of 20 μg mL^−1^ and after 6 h incubation, the cells were irradiated with a NIR laser (808 nm, 1.0 W cm^−2^) for 5 min followed by 24 h incubation. The results demonstrated that mPEG-Pep-IDOi/ICG could enhance killing efficacy through CCK-8 assay, induce ICD of tumor cells and upregulate CD80 and CD86, sustaining the DC maturation. Regarding in vivo tests, C57BL/6 mice were inoculated with B16-F10 cells and treated mPEG-Pep-IDOi/ICG with a concentration of ICG and IDO of 4 and 5 mg kg^−1^, respectively. After 4h, mice were irradiated under the same setting used in the in vitro studies. The authors reported an increased expression of CD80 and CD86, meaning an increase in DC maturation, a tumor-inhibiting effect (tumor volume ≈ 0 mm^3^), and higher levels of INF-γ, TNF-α and IL-6. The treatment promoted synergetic antitumor responses to efficiently reduce tumor growth.

Zang et al. [106] developed a pH-responsive photo-immune nano-booster (NC@Ce6) for amplifying photodynamic-immunotherapy. The NC@Ce6 comprises the aPDL1-PC constructed by reversibly grafting aPDL1 with 3,4-dihydroxybenzaldehyde (Protocatechualdehyde, PC) via amine-aldehyde condensation, the phenylboronic acid (PBA) modified branched polyethyleneimine (PEI-PBA), and the loaded Ce6. The NC@Ce6 nanoparticles were used at a concentration of 7.5 μg mL^−1^ against B16F10 melanoma cells and incubated for 2.5 h. After, the cells were irradiated with a laser (650 nm, 20 mW cm^−2^) for 2.5 min and further incubated for 2 h. The nanocomplex induced and increased CRT exposure on the surface of tumor cells, leading to an apoptosis degree of approximately 34% (Annexin V-FITC), 2.1-fold higher than PBS control, and increased DC maturation. Regarding in vivo tests, B16F10 tumor-bearing mice were then injected with NC@Ce6 ([Ce6] = 2.5 mg kg^−1^). After 2 h incubation, mice were irradiated with a laser (650 nm, 100 mW cm^−2^) for 10 min. NC@Ce6 under laser irradiation group had the best tumor inhibition rate (78% tumor weight), the tumor infiltrating T cells values was approximately 49% compared to around 20% measured in the PBS group, also results demonstrated DC maturation. Globally, this treatment has revealed to be able to trigger a strong immune response.

Wu et al. [101] reported IR780 molecules and Tyrosinase-related protein 2 (TRP-2) peptide encapsulation in the hydrophobic shell, and hydrophilic interior of TAT peptide functionalized liposomes to form _T_LipIT NPs with a spherical structure. _T_LipIT present an average size of 180.4 ± 10.2 nm and hydrodynamic size of 225.9 ± 5.1 nm. After irradiating for 5 min, the liposomes could increase PBS temperature by 16 °C. _T_LipIT were used at a concentration of 100 μg mL^−1^ under laser irradiation (808 nm, 0.75 W cm^−2^) for 5 min against B16F10 cells. This treatment could induce a higher secretion of TNF-α and INF-γ due to the natural release of TRP-2 peptide. The irradiation also led to approximately 12% of early apoptosis and 37% of late apoptosis (Annexin V-FITC). _T_LipIT were internalized into neutrophils (NEs) (_T_LipIT/NEs) (100 μg mL^−1^) and used to treat B16F10 tumor-bearing mice treated under NIR irradiation (808 nm, 0.75 W cm^−2^, 5 min). Regarding in vivo tests, _T_LipIT/NEs under laser irradiation had the best local tumor suppression, the CD80+ and CD86+ mature DCs frequency was ≈33%, significantly higher than the ≈ 16% measured without NIR irradiation, and the ≈11% registered when PBS only was used. CD4+ and CD8+ T lymphocyte frequencies were approximately 49 and 33%, respectively. These results demonstrate that _T_LipIT/NEs nanoparticles prompt the best immune response and tumor damage without causing pathological changes under NIR irradiation.

Black phosphorus (BP) is a nanosheet with a 2D puckered honeycomb structure which has also been studied for photoimmunotherapy of skin cancer [93,94]. Wan et al. [93] PEGylated black phosphorus nanosheets (BP-PEG NSs) and combined those with imiquimod. BP lateral size was around 100 nm with a height between 8 and 12 nm. The zeta potential increased after coating with PEG (from −18.8 to ≈ −8.3 mV). BP-PEG with imiquimod, at a concentration of 10 μg mL^−1^, was used to treat B16 melanoma cells and combined with NIR laser irradiation (808 nm, 3.2 W cm^−2^) for 10 min after 6 h incubation followed by another 6 h. BP-PEG with imiquimod combined with laser irradiation could promote the secretion of TNF-α, IL-6, and IL-12 cytokines and increase the maturation of DCs by ≈31%, demonstrating that a combination of PTT with immunotherapy can induce a significant immune response. Regarding in vivo tests, C57BL/6 mice treated with BP-PEG and imiquimod (0.5 mg kg^−1^, 25 μL) under laser irradiation (24 h post-injection) also increased the secretion of TNF-α, IL-6 and IL-12 cytokines. These results showed that this treatment can enhance the immune response and promote an anti-tumor effect. DC maturation also increased by ≈46%, implicating that tumor ablation recruits DCs to the tumor site. BP-PEG + R837 group + irradiation showed the highest percentage (≈18%) of CD8+ T cells. These results demonstrated that BP-PEG + R837 photoimmunotherapy can induce a potent response. Li et al. [94] showed that BP nanoparticles coated with phenylalanine-lysine-phenylalanine (FKF) tripeptide-modified antigen epitopes (FKF-OVAp@BP) (15.9 μg per mouse), combined with NIR irradiation 6 h after immunization (808 nm, 0.5 W cm^−2^) for 5 min had a potent immune response and anti-tumor protection effect, leading to 100% survival over 60 days in C57BL/6 mice injected with B16-OVA cells. There was also an increase in the DC maturation and in effector and central memory of CD8+ T cells, meaning that this treatment enhanced T-cell immunity memory.

## 4. Biocompatibility, Safety, and Translational Challenges of Nanomaterials for Skin Cancer Photoimmunotherapy

The use of nanomaterials for biomedical applications implies great care with biocompatibility and safety issues. Each nanomaterial has unique properties that influence these aspects, such as size and surface area, particle shape, surface charge, composition, structure, and degree of exfoliation. Such characteristics have a great impact on how the human body responds, distributes, and eliminates nanomaterials. The US-FDA (United States—Food and Drug Administration) requires biocompatibility and safety tests to be performed within the context in which the materials will be used [110].

Nanomaterials biocompatibility should be firstly tested in vitro and afterwards in vivo. In vitro, aluminum hydroxide nanoparticles showed good biocompatibility and low cytotoxicity towards B16F10 cells, even at high concentrations of 5 μg mL^−1^ [92]. Chitosan nanoparticles showed high phototoxicity and low dark cytotoxicity towards B16F10 cells. The intracellular ROS production was positively correlated with the dose of the nanoparticles and was responsible for cell killing [95]. Black phosphorus revealed excellent biocompatibility, even at high concentrations (100 μg mL^−1^) towards B16F10 cells [93]. FKF-peptide coating led to biocompatibility increases of 20 and 60% in bone marrow-derived dendritic cells (BMDCs) and splenocytes, respectively [94]. Liposome nanoparticles showed little effect on the viability of B16F10 cells. The toxicity under laser irradiation was attributed to their photothermal and photodynamic properties [101]. Polydopamine nanoparticles at concentrations between 10 to 300 μg mL^−1^ were revealed to be nontoxic against B16F10 cells [107]. Gold nanoparticles revealed no cytotoxicity with concentrations between 0 and 100 μM towards B16F10 cells [97]. B16F10 cells treated with mPEG-Pep-IDOi/ICG at 15 μg mL^−1^ presented 80% cell viability [104]. In the case of NC@Ce6 (0.937 and 7.5 μg mL^−1^) at pH 7.4, the viability of the cancer cells was not affected without the laser irradiation being performed [106].

Regarding the biocompatibility and safety tests performed in vivo, aluminum hydroxide showed no substantial toxicity to the organs, as demonstrated by tumor histological analysis [92]. Chitosan nanoparticles led to inflammation at the administration site but no histological damage, showing an absence of toxicity [95]. The administration of black phosphorus to B16 tumor-bearing mice led to no decrease in body weight and no toxicity to the main organs [93,94]. Micelles and polydopamine revealed minimal toxicity according to histological analysis [102,103,107]. L-Ce6 MNs ([Ce6] = 0.12 mg kg^−1^) caused no significant differences in body weight and main organ indices and no obvious inflammatory infiltration or other pathological changes. Minor trauma to the skin at the administration site was observed; however, the skin barrier damage was transient and reversible [100]. Gold nanoparticles were revealed to be nontoxic and have good biocompatibility in vivo once there were no pathological changes identified in any organs and no obvious inflammation observed [98]. Silicon dioxide showed some accumulation in mice’s liver and kidneys, according to the hematological assay red, without major alterations on white blood cells and platelet counts. There was no significant decrease in body weight, supporting the non-toxicity of these nanoparticles [109]. Despite the results presented above, the literature still lacks more extensive studies regarding the safety and biocompatibility of nanoparticles used for skin cancer photoimmunotherapy both in vitro and in vivo, particularly regarding long-term toxicity and bioaccumulation studies and tests performed in more complex in vivo models.

There are some translational challenges regarding nanomaterial use in clinics. These challenges can be divided into (i) biological, the internalization and degradation of nanoparticles are important for clinical validation; (ii) scale-up, some processes can be less efficient on a large scale (e.g., ultrafiltration); (iii) storage, nanomedicines have rigorous storage conditions that are often too expensive for low-income countries to afford; (iv) biosafety, when scaling up some changes in the production of nanoparticles can change the quality and safety of the products; (v) economic, the costs of production are often high, and the relation between nanomaterials costs and consumers purchasing power may lead to product failure in the market [111,112,113].

At the moment, there is a clinical trial using ASP-1929 combined with anti-PD1 therapy for photoimmunotherapy of recurrent/metastatic head and neck squamous cell carcinoma or for advanced/metastatic cutaneous squamous cell carcinoma. Currently, there are only a few nanomaterials/molecules patented that can be used for skin cancer photoimmunotherapy. The patents for those materials are US10537641B2, which protects an IR700 molecule conjugated to an antibody that binds HER1; EP3148583B1, which protects an agent comprising a phthalocyanine dye IRDye^®^ 700DX conjugated to a probe that has a molecular weight below 10 kDa; WO2022239766A1, that protects an antibody that binds to CADM1 (Cell Adhesion Molecule 1) on the cell surface; and WO2020163656A1, that protects a formulation that contains a DR6 peptide.

## 5. Conclusions and Outlook

Nanomaterials tested so far for photoimmunotherapy belong to four major classes, namely metallic, polymeric, lipid-based, and 2D nanoparticles. The increased secretion of pro-inflammatory cytokines and DC maturation proved that the treatments could contradict the tumoral microenvironment and create a pro-inflammatory response at tumor site. Additionally, photodynamic/photothermal therapy combination with immunotherapy allowed a reduction of primary tumors size, not seen when only immunotherapy was applied.

All studies used lasers to irradiate the cells. The most common nanoparticles used were gold, silicon dioxide, micelles and black phosphorus. Black phosphorus studies in vitro revealed a 49% cancer death after treatment. Tumor volumes decreased by 3 or 10 fold, being PEGylated black phosphorus loaded with imiquimod (808 nm, 3.2 W cm^−2^, 10 min) [94] responsible for the best results. For gold nanoparticles, in vitro cancer cell death ranged from 88 to 60%. The most effective modifications were BSA and HA surface adsorption, while the most effective substances combined were M2pep, R837, and simvastatin. All those gold-nanocomposites led to tumor volume reductions of around 10-fold compared with controls [96,97,99]. For silicon dioxide nanoparticles, modification by CuS loading inside the pores, PDMAEMA surface adsorption, and integration with the plasmid encoding IL-12 gene, combined with laser irradiation (1064 nm, 0.65 W cm^−2^, 5 min) revealed the best results. In vitro cancer cell death was over 20%, and tumor volume decreased ≈ 3-fold compared with control at day 18 [108]. Micelles have been successfully loaded with chloroquine + IR780 or imiquimod. When using micelles, there was not any physicochemical modification performed, in vitro cell death was approximately 80%, and in vivo complete tumor eradication was achieved at day 12 [102] or day 25 [103] after treatment with laser irradiation (808 nm, 1.0 W cm^−2^, 5 min). Overall, micelles have produced the best results from the most common nanomaterials reported. Other nanomaterials induced complete tumor eradication in vivo, namely mPEG-Pep-IDOi/ICG NPs, and aluminum hydroxide (Table 1). However, more studies need to be performed for each particular nanomaterial class mentioned to confirm those effects. Almost all studies have revealed that a combination of photodynamic/photothermal therapy and immunotherapy produces synergetic effects, yielding higher tumor eradication and secretion of pro-inflammatory cytokines.

Finally, results are very promising so far due to the demonstration of synergism between both photodynamic/photothermal therapy and immunotherapy and to the reduction of metastatic cancer, showing that the new approach of using combined photoimmunotherapy is ideal for solving some of the existing problems related to conventional cancer therapy methods. Photoimmunotherapy has revealed better efficacy in eradicating primary tumors, preventing distant tumors, and inhibiting metastasis compared with immunotherapy or photodynamic/photothermal therapy alone. It is important to notice that detailed toxicity studies are still lacking in vivo, and the dissection of the underlying molecular mechanisms requires further investigation. Therefore, for this new promising approach to reach its potential, further research on the topic is paramount.

## Figures and Tables

**Figure 1 biomedicines-11-01292-f001:**
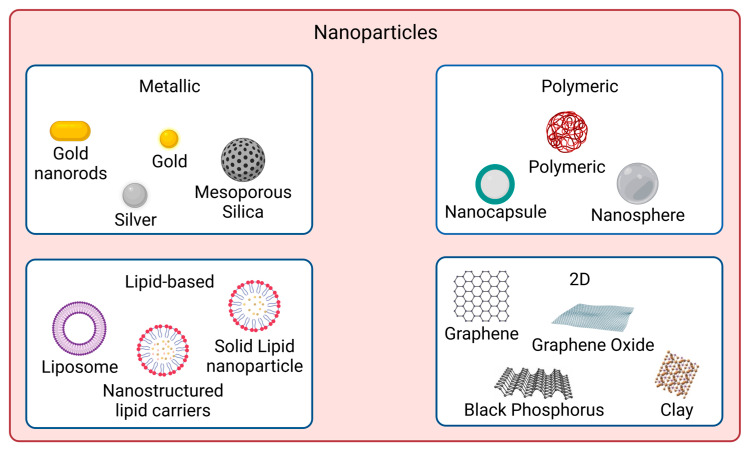
Nanomaterial types used for skin cancer photoimmunotherapy. The main categories are metallic, polymeric, lipid-based, and 2D nanoparticles. Created with BioRender.com.

**Figure 2 biomedicines-11-01292-f002:**
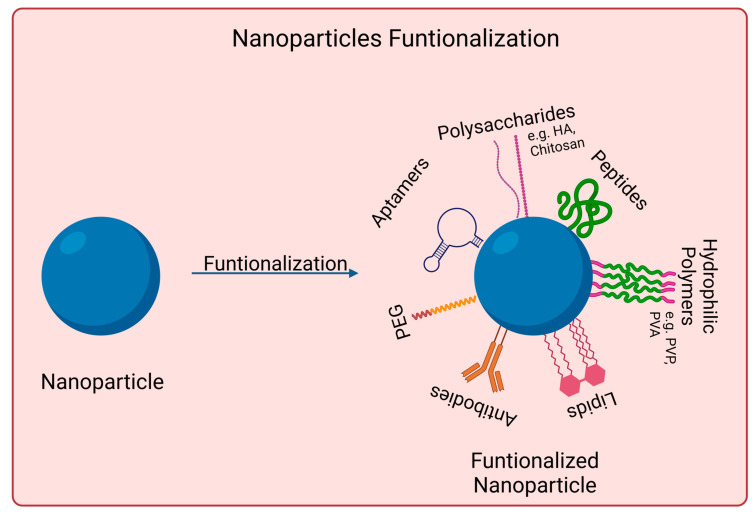
Surface functionalization strategies used for skin cancer photoimmunotherapy. Functionalization with PEG (polyethylene glycol), polysaccharides, lipids, polymers, aptamers, peptides, and antibodies. Created in Biorender.com. Abbreviations: HA, hyaluronic acid; PVP, poly-(vinylpyrrolidone); PVA, Polyvinyl alcohol. Created with BioRender.com.

**Figure 3 biomedicines-11-01292-f003:**
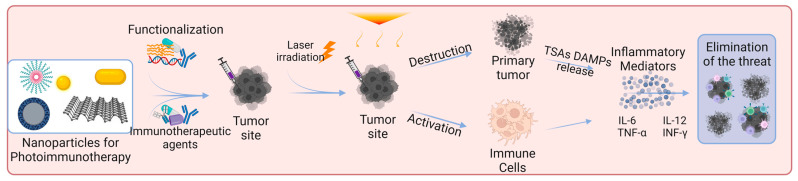
Skin cancer photoimmunotherapy using functionalized nanoparticles. The nanoparticles can exert a direct effect on cells by destroying primary tumors and activating immune cells. Photoimmunotherapy can induce an inflammatory response and increase the release of pro-inflammatory cytokines (e.g., TNF-α, IFN-γ, IL-6, IL-12). Nanoparticles were modified using specific drugs, polymers, or antibodies to induce a desired immune response. Abbreviations: TSAs, tumor-specific antigens; DAMPs, damage-associated molecular patterns; IL-6, interleukin-6; IL-12, interleukin-12; TNF-α, tumor necrosis factor alpha; INF-γ, interferon gamma. Created with BioRender.com.

**Figure 4 biomedicines-11-01292-f004:**
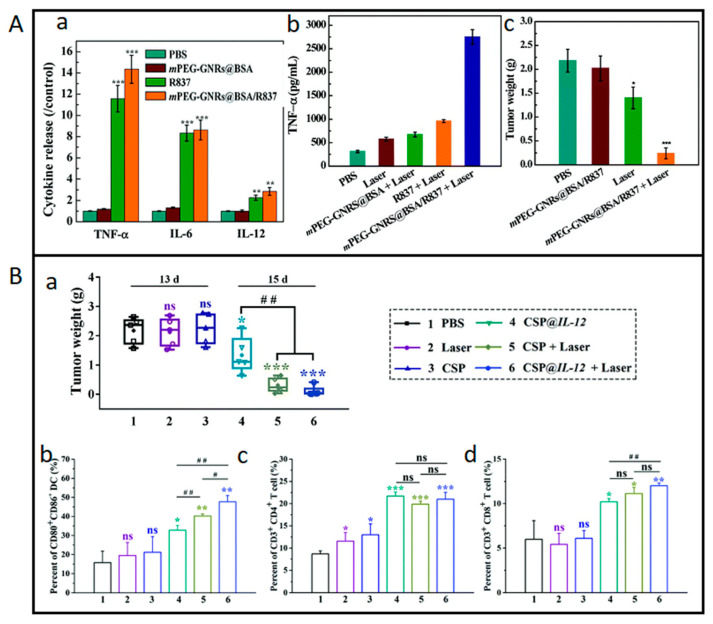
Examples of skin cancer photo-immunotherapeutic modulation using nanomaterials. (**A**) BSA-bioinspired gold nanorods (GNRs) decorated with PEG and loaded with R837 administered to B16F10 tumor-bearing mice followed by NIR laser irradiation (1064 nm, 1.0 W cm^−2^, 10 min), mPEG-GNRs@BSA/R837 + Laser group; (**a**) Cytokine levels of TNF-α, IL-6, and IL-12 in the serum of mice 3 days after various treatments; (**b**) Cytokine secretion of TNF-α from macrophages stimulated by treated B16-F10 tumor after various treatments; (**c**) Weight of primary tumor 15 days after various treatments. * *p* < 0.05, ** *p* < 0.01, *** *p* < 0.001. Used with permission of Royal Society of Chemistry, from [97]; permission conveyed through Copyright Clearance Center, Inc. (**B**) PDMAEMA polycation on the outside of SiO_2_ surface integrated with the plasmid encoding IL-12 gene (1064 nm, 0.65 W cm^−2^, 5 min), CSP@IL-12 + Laser group; (**a**) Average weights of the excised tumors after different treatments as indicated (* *p* < 0.05, ** *p* < 0.01, *** *p* < 0.001, compared to the PBS group; # *p* < 0.05, ## *p* < 0.01, pairwise comparison; n = 5); (**b**) The percentages of CD80+/CD86+ DC cells in the tumor-draining lymph nodes (gated on CD11c+).; (**c**) The percentages of CD3+/CD4+ T cells in the spleen; (**d**) The percentages of CD3+/CD8+ T cells. (* *p* < 0.05, ** *p* < 0.01, *** *p* < 0.001, compared to the PBS group; # *p* < 0.05, ## *p* < 0.01, pairwise comparison; n = 3). Data are presented as mean ± SD. Used with permission of Royal Society of Chemistry, from [108]; permission conveyed through Copyright Clearance Center, Inc.

**Figure 5 biomedicines-11-01292-f005:**
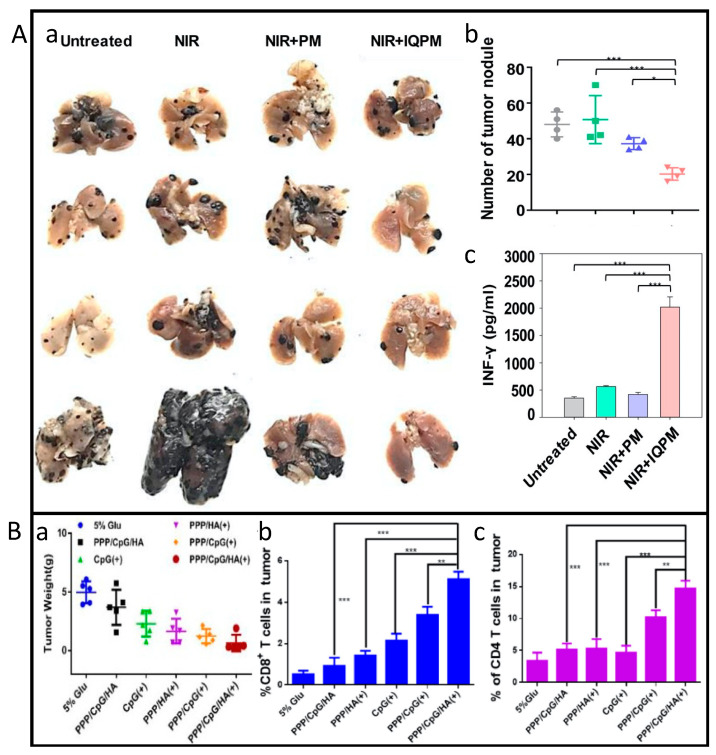
In vivo studies on skin cancer photo-immunomodulation effects induced by nanomaterials. (**A**) Imiquimod encapsulated in amphiphilic peptide-based micelles (IQPM) applied to B16F10 tumor-bearing mice together with NIR laser irradiation (808 nm, 1.5 W cm^−1^, 5 min); (**a**) Photographs of lung tissues taken 21 days after treatments; (**b**) Quantification of average number of tumor nodules in lung tissues (n = 4, one-way ANOVA and Student–Newman–Keuls test, * *p* < 0.05; *** *p* < 0.001); (**c**) IFN-𝛾 secretion from splenocytes of treated mice (n = 3, one-way ANOVA and Student–Newman–Keuls test, *** *p* < 0.001). Reproduced with permission from [103] © 2023 Wiley-VCH GmbH. (**B**) PPP/CpG/HA nanoplatform applied to B16-F10 tumor mice model together with laser irradiation (808 nm, 1.5 W cm^−2^, 5 min); (**a**) Weight of isolated tumors after treatment (mean ± SD, n = 5); (**b**) Frequency of infiltrating CD8+ T cells in tumor; (**c**) infiltrating CD8+ T cells in spleens. ∗∗, and ∗∗∗ indicate *p* < 0.05, *p* < 0.01 and *p* < 0.001, respectively. Reprinted from [107], Copyright @ 2023, with permission from Elsevier.

**Table 1 biomedicines-11-01292-t001:** Nanomaterials for skin cancer photoimmunotherapy. Their properties, composition, and biological effects.

Nanomaterial	Physicochemical Modifications	Loaded Substance	Particle Size	In Vitro Studies		In Vivo Studies		Ref.
				Parameters	Results	Parameters	Results	
**Aluminum hydroxide**	BSA surface adsorption	Chlorin e6	25.25 ± 2.1 nm	B16F10 cells [Al-BSA-Ce6] = 0.1 μg mL^−1^ I: 660 nm, 0.8 W cm^−2^, 5 min	95% cell death (MTT assay) ↑CD80 ↑TNF-α, IL-12p70, IL-1β	C57BL/6 mice subcutaneously injected with B16F10 cells [Al-BSA-Ce6] = 5 mg kg^−1^ I: 660 nm, 0.8 W cm^−2^, 5 min	Tumor volume of 0 mm^3^ at day 7 ≈63% mice survived 100 days ↑T cells tumor infiltration ↑TNF-α and INF-γ production	[92]
**Black Phosphorus**	PEG electrostatic adsorption	Imiquimod	≈120 nm	B16 cells [BP-PEG] = 10 μg mL^−1^ I: 808 nm, 3.2 W cm^−2^, 10 min	45% cell viability↓ (MTT assay) ↑TNF-α, IL-6, IL-12 DCs↑ 30.8%	C57BL/6 mice subcutaneously injected with B16 [BP] = 0.5 mg kg^−1^, 25 μL [R837] = 0.35 mg kg^−1^, 25 μL I: 808 nm, 3.2 W cm^−2^, 3 min	≈10-fold tumor vol.↓ DCs↑ 45.5% ↑TNF-α, IL-6, IL-12	[93]
**Black Phosphorous**	N/A	FKF-OVAp	≈500 × 23 nm	N/A	N/A	C57BL/6 mice subcutaneously injected with B16-OVA [FKF-OVAp] = 10 nmol per mouse [BPs] = 15.9 μg per mouse I: 808 nm, 0.5 W cm^−2^, 5 min	≈3-fold tumor vol.↓ 100% survival over 60 days ↑DC activation ↑CD8+ T cells effector and central memory	[94]
**Chitosan**	Cross-linking, sodium tripolyphosphate	IDO	220 nm	B16 cells [ICG-NP] = 30 μg mL^−1^ I: 808 nm, 0.35 W cm^−2^, 5 min	≈0% cell viability (CCK8 assay) ≈85% DC frequency	C57BL/6 mice subcutaneously injected with B16F10 Drug loading: 4 and 35 μg per microneedle patch of ICG and 1-MT, respectively I: 808 nm, 0.35 W cm^−2^, 5 min 3 cycles at interval of 2 days	Tumor volume of ≈0 mm^3^ 80% survival rate without recurrence after 120 days ≈55% DC maturation level ↑CD8+ T cells in distant tumor ↑TNF-α, IL-12p70, IL-6	[95]
**Gold**	HA surface adsorption	M2pep	64.6 nm	B16F10 cells [HA-AuNR/M-M2pep] = 20 μg mL^−1^ I: 808 nm, 1.5 W cm^−2^, 2 min	40% cell viability (CCK8 assay) 35.1 ± 1.8% apoptosis (Annecin V-FITC) ↑CRT-positive cells and HMGB1 release	C57BL/6 mice subcutaneously injected with B16F10 [AuNR] + [M2pep] = 10 + 12 mg kg^−1^ I: 808 nm, 1.5 W cm^−2^, 2 min	≈10-fold tumor vol.↓ 67% survival rate at 45 days 3.7-fold↑ CD8+ T cells INF-γ, TNF-α↑ ≈ 6-fold	[96]
**Gold**	BSA surface adsorption	R837	122.1 ± 11.6 nm	B16-F10 cells [Au] = 11.5 μg mL^−1^ I: 1064 nm, 1.0 W cm^−2^, 10 min	Cell viability↓ to ≈27% (MTS assay) HSP70/β-Actin release ≈ 0	C57BL/6 mice subcutaneously injected with B16F10 [Au] = 300 μg mL^−1^ I: 1064 nm, 1.0 W cm^−2^, 10 min	≈10-fold tumor vol.↓ TNF-α, IL-6, IL-12 ≈ 14, 9, 3 times↑ than PBS, respectively ↑CD8^+^ T cells infiltration	[97]
**Gold**	Gold nanoparticles retained in extracellular vesicles with tumor antigens (AuNP@B16F10)	Tumor antigens (AuNP@B16F10)	40 nm	N/A	N/A	Murine melanoma model subcutaneously injected with AuNP@DC_B16F10_ [Au] = 1.35 mg kg^−1^ I: 808 nm, 2.0 W cm^−2^, 1 min Cycle: 3 times, 3 days interval	69% tumor volume↓ 50% tumor-free mice at day 19 Distant tumor inhibition ↑CD3^+^ and CD8^+^ T cells infiltration ↑INF-γ, TNF-α, IL-6	[98]
**Gold**	N/A	SV	50 nm	B16-F10 cells [Au] = 60 μg mL^−1^ I: 808 nm, 1.5 W cm^−2^, 2 min	≈12% cell viability (MTT assay) CRT expression ≈3.5-fold↑ ↑mature DCs frequency	B16F10-bearing C57 mice I: 808 nm, 1.5 W cm^−2^, 2 min	≈10-fold tumor vol.↓ ↑DC maturation CD4^+^ and CD8^+^ T cells proliferation	[99]
**Hyaluronic acid**	Self-assembly of Ce6/α-linoleic acid (L-Ce6 NAs (nano-assemblies)) Fast dissolving L-Ce6 NAs in oligo-HA and micro-molding of microneedles (tips enriched with 3 µg Ce6)	Ce6	≈86 nm	B16F10 cells [Ce6] = 400 μM I: 660 nm, 200 mW cm^−2^, 5 min	CRT fluorescence ↑2-fold ATP secretion ≈ 2.5 nM ↑HMGB1 release	C57BL/6 mice subcutaneously injected with B16F10 [Ce6] = 0.12 mg kg^−1^ I: 660 nm, 200 mW cm^−2^, 4 min	≈3-fold tumor vol.↓ ↑CD4+ and CD8+ ≈3 and 4-fold	[100]
**Liposomes**	N/A	TRP-2	180.4 ± 10.2 nm	B16F10 cells [_T_LipIT NPs] = 100 μg mL^−1^ I: 808 nm, 0.75 W cm^−2^, 5 min	≈12% early apoptosis (Annexin V-FITC) 37% late apoptosis ↑TNF-α, INF-γ	C57BL/6 mice subcutaneously injected with B16F10 [_T_LipIT/NEs] = 100 μg mL^−1^ I: 808 nm, 0.75 W cm^−2^, 5 min	≈10-fold tumor vol.↓ ≈33% CD80^+^ and CD86^+^ mature DCs frequency ≈49 and ≈33% CD4^+^ and CD8^+^ T lymphocytes frequency	[101]
**Micelles**	N/A	CQ IR780	80−90 nm	B16 cells [C/I-Mil] = 4 μg mL^−1^ I: 808 nm, 1.0 W cm^−2^, 5 min	20% cell viability (CCK-8 assay) Cell membrane integrity destroyed Phagocytic index ↑3.0-fold	C57BL/6 mice subcutaneously injected with B16F10 [C/I-Mil] + [CQ/Mil] = 4 μg mL^−1^ + 20 μg/patch I: 808 nm, 1.0 W cm^−2^, 5 min	Primary tumor suppression: 0 mm^3^ 50% survived at least 40 days Distant tumor volume ↓3.4-fold	[102]
**Micelles**	N/A	Imiquimod	72.0 ± 18.0 nm	N/A	N/A	C57BL mice subcutaneously injected with B16 cells [IQPM] = 5 mg kg^−1^ I: 808 nm, 1.5 W cm^−1^, 5 min	Primary tumor suppression: 0 mm^3^ CD8^+^ and CD4^+^ T cells ↑9.3- and 10.3-fold 2.4-fold↓ metastasis	[103]
**mPEG-Pep-IDOi/ICG NPs**	N/A	N/A	140 nm	B16-F10 cells [ICG] = 20 μg mL^−1^ I: 808 nm, 1.0 W cm^−2^, 5 min	≈0% cell viability (CCK-8 assay) Induced ICD of tumor cells ≈70% CD80 and CD86↑	C57BL/6 mice subcutaneously injected with B16-F10 [ICG] = 4 mg kg^−1^ [IDOi] = 5 mg kg^−1^ I: 808 nm, 1.0 W cm^−2^, 5 min	Primary tumor suppression: 0 mm^3^ CD80+ and CD86+ ↑13.5 and 12.3% ↑INF-γ, TNF-α, IL-6	[104]
**PBE**	N/A	RSL-3	<100 nm	B16-F10 cells [RSL-3] = 0.5 μg mL^−1^ [INF-γ] = 100 ng mL^−1^ I: 671 nm, 100 mW cm^−2^, 1 min	30% mature DC CTR expression ↑5.0-fold	C57BL/6 mice subcutaneously injected with B16-F10 [RSL-3] = 0.5 μg mL^−1^ [INF-γ] = 100 ng mL^−1^ I: 671 nm, 150 mW cm^−2^, 2 min	≈2-fold tumor vol.↓ ≈50% survival rate ≈30% mature DC cells INF-γ secretion ↑ 4-fold	[105]
**PEI-PBA**	PEG surface adsorption	Ce6 aPDL1	117 ± 4.0 nm	B16F10 cells [NC@Ce6-pH 6.0] = 7.5 μg mL^−1^ I: 650 nm, 20 mW cm^−2^, 2.5 min	89% CRT rate ≈34% apoptosis (Annexin V-FITC) DC maturation	B16F10 tumor-bearing mice [Ce6] = 2.5 mg kg^−1^ I: 650 nm, 100 mW cm^−2^, 10 min	78% tumor inhibition rate ≈49% tumor infiltrating T cells DC maturation	[106]
**Polydopamine**	PEI surface adsorption	CpG oligodeoxynucleotides	140 nm	B16F10 cells [PPP/CpG/HA] = 200 μg mL^−1^ I: 808 nm, 2.0 W cm^−2^, 5 min	≈5% cell viability (MTT assay) ≈60% apoptosis (Annexin V-FITC) ≈60% CD80+ DC ≈50% CD86+ DC	C57BL/6 mice subcutaneously injected with B16F10 [PPP/CpG/HA] = 0.75 mg kg^−1^ I: 808 nm, 1.5 W cm^−2^, 5 min	≈20-fold tumor vol.↓ Largest apoptotic cell area CD80+ DC ≈ 3% CD86+ DC ≈ 4.5%	[107]
**Silicon Dioxide**	CuS loaded inside the pores PDMAEMA surface adsorption	IL-12 gene	157 nm	B16F10 cells [CSP] = 34.5 μg mL^−1^ I: 1064 nm, 0.65 W cm^−2^, 5 min	<20% cell viability (CCK-8 assay) 89% apoptotic cells (Annexin V-FITC) ↑CRT expression 66% DCs maturation	B16F10-bearing C57BL/6 mice [CSP] = 172.4 μg per mouse I: 1064 nm, 0.65 W cm^−2^, 5 min	≈3-fold tumor vol.↓ Prolonged survival DC maturation level: ≈48% 21% CD4^+^ and 12% CD8^+^ T populations	[108]
**Silicon Dioxide**	Chemical synthesis of UCNP@m-SiO_2_@liposome NPs	Ce6 and BSO	≈50 nm	B16/F10 cells [UCB] = 100 μg mL^−1^ I: 980 nm, 0.7 W cm^−2^, 10 min	≈29% cell viability (CCK-8 assay) ≈39% apoptosis rate (Western Blot) ↑TNF-α, IL-6, INF-γ	C57BL/6 mice subcutaneously injected with B16F10 [UCB] = 0.8 mg per mouse I: 980 nm, 0.7 W cm^−2^, 20 min	≈3-fold tumor vol.↓ ↑IL-12p40, INF-γ	[109]

Abbreviations: ↑, increase; ↓, decrease; aPDL1, anti-programmed death-ligand 1; BSA, bovine serum albumin; B16 cells, B16 murine melanoma cell line; BSO, buthionine sulfoximine B16F10 cells, B16F10 murine melanoma cell line; B16-F10 cells, B16–F10 murine melanoma cell line; B16/F10 cells, B16/F10 murine melanoma cell line; CCK-8, cell counting kit-8; Ce6, chlorin e6; CQ, chloroquine; DC, dendritic cells; HA, hyaluronic acid; IDO, indoleamine 2,3-dioxygenase; M2pep, M2 macrophage-targeting peptide; MTT, 3-(4,5-dimethylthiazol-2-yl)-2,5-diphenyl-2H-tetrazolium bromide; N/A, not applicable; PDMAEMA, poly((2-di- methylamino)ethyl methacrylate); PEG, polyethylene glycol; PEI, polyethyleneimine; R837, imiquimod; SV, simvastatin; TRP-2, tyrosinase-related protein 2.

## Data Availability

No new data were created or analyzed in this study. Data sharing is not applicable to this article.

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
