# Peer review of "Nanomaterials for Skin Cancer Photoimmunotherapy"

_biomedicines, 2023, doi:10.3390/biomedicines11051292_

Round 1
Reviewer 1 Report
The article discusses the use of different types of nanomaterials in photoimmunotherapy to treat cancer. Metallic, polymeric, lipid-based, and 2D nanoparticles have been tested, and results have shown that the treatments can create a pro-inflammatory response at the tumor site. The most common nanoparticles used were gold, silicon dioxide, micelles, and black phosphorus. Studies have shown results in reducing primary tumors, and combination with immunotherapy has been effective. Micelles have produced the suitable results, but more research is needed to confirm the effects of each nanomaterial class. The article concludes that the combination of phototherapy and immunotherapy is ideal for solving some of the problems related to conventional cancer therapy methods. However, further research is needed to fully understand the underlying mechanisms and potential toxicity.
I have carefully reviewed your manuscript and unfortunately, I must inform you that I cannot recommend it for publication.
My assessment of the paper suggests that it lacks novelty and the level of impact required for publication in this journal. Additionally, the article would benefit from further experimental and theoretical investigation to strengthen the scientific merit.
I understand the effort and time that has been invested in this research, but at this stage, I am unable to recommend it for publication.
Author Response
Manuscript ID: biomedicines-2338753
Title: Nanomaterials for Skin Cancer Photoimmunotherapy
Authors: Carlota M. Relvas, Susana G. Santos, Maria J. Oliveira, Fernão D. Magalhães, Artur M. Pinto
We sincerely appreciate all comments and suggestions received and have now carefully replied to all reviewer’s comments point by point and edited the manuscript text accordingly. We believe that the quality and impact of the work has been significantly improved thanks to it and hope that now you find it suitable for publication in Biomedicines Journal. All changes in the manuscript text have been highlighted in green.
Reviewer 1 comments:
The article discusses the use of different types of nanomaterials in photoimmunotherapy to treat cancer. Metallic, polymeric, lipid-based, and 2D nanoparticles have been tested, and results have shown that the treatments can create a pro-inflammatory response at the tumor site. The most common nanoparticles used were gold, silicon dioxide, micelles, and black phosphorus. Studies have shown results in reducing primary tumors, and combination with immunotherapy has been effective. Micelles have produced the suitable results, but more research is needed to confirm the effects of each nanomaterial class. The article concludes that the combination of phototherapy and immunotherapy is ideal for solving some of the problems related to conventional cancer therapy methods. However, further research is needed to fully understand the underlying mechanisms and potential toxicity.
I have carefully reviewed your manuscript and unfortunately, I must inform you that I cannot recommend it for publication.
My assessment of the paper suggests that it lacks novelty and the level of impact required for publication in this journal. Additionally, the article would benefit from further experimental and theoretical investigation to strengthen the scientific merit.
I understand the effort and time that has been invested in this research, but at this stage, I am unable to recommend it for publication.
Answer: Concerning your comments on the need for further research, please note that this manuscript is a review of the current state of the art and not a report on original research work. We agree that more studies are needed in the areas you pointed out, and this has been highlighted in the manuscript. The text has been improved according to all reviewers’ comments and changes in the manuscript can be found highlighted in green.
Reviewer 2 Report
The review manuscript entitled "Nanomaterials for Skin Cancer Photoimmunotherapy" was well written and organized with well described case studies. The manuscript needs some major changes before acceptable for publication.
As per the abstract outline, the mechanisms of the nanomaterials not discussed in-detail. Recent advancements has been developed with nanotheranostics for melanoma. Write a note on this.
Add a note on statistical demographic details on the skin cancer and the treatment burden to the economy.
What are the scaleup challenges for the development of nano carriers would be the added advantage to the formulation scientist.
Write the recent clinical trails and patents of the same.
Explain the different modalities (small and large molecules) encapsulation techniques for skin cancer tretament.
Author Response
Manuscript ID: biomedicines-2338753
Title: Nanomaterials for Skin Cancer Photoimmunotherapy
Authors: Carlota M. Relvas, Susana G. Santos, Maria J. Oliveira, Fernão D. Magalhães, Artur M. Pinto
We sincerely appreciate all comments and suggestions received and have now carefully replied to all reviewer’s comments point by point and edited the manuscript text accordingly. We believe that the quality and impact of the work has been significantly improved thanks to it and hope that now you find it suitable for publication in Biomedicines Journal. All changes in the manuscript text have been highlighted in green.
Reviewer 2 Comments:
The review manuscript entitled "Nanomaterials for Skin Cancer Photoimmunotherapy" was well written and organized with well described case studies. The manuscript needs some major changes before acceptable for publication.
1. As per the abstract outline, the mechanisms of the nanomaterials not discussed in-detail. Recent advancements has been developed with nanotheranostics for melanoma. Write a note on this.
Answer: We appreciate the reviewer suggestion and had, therefore, added a paragraph regarding nanotheranostics applications for melanoma in Introduction, page 2, line 56: “More recently, new approaches, such as the use of metallic, polymeric, and lipid-based nanotheranostics, emerged in this field to facilitate the treatment and the diagnosis of this disease. These platforms may serve as contrast agents with therapeutic properties (e.g., reactive oxygen species production). Moreover, the versatility of their formulations allows incorporation of other materials (e.g., gold) and/or drugs (e.g., parvifloron D), which act synergistically to impair the development of solid and superficial tumors, namely at early stages [16,17]. All above mentioned new therapies and nanomaterials mechanisms of action will be discussed in more detail in the next sections (e.g., Figure 3).”
2. Add a note on statistical demographic details on the skin cancer and the treatment burden to the economy.
Answer: A note has been added regarding statistical demographic details on skin cancer – Introduction, page 1, line 29: “Concerning its geographic distribution, North America is estimated to represent the majority of the cases (45.4%), followed by Europe (28.4%), Central and South America (9.5%), Australia and New Zealand (5.8%). The highest incidence of this disease occurs in Europe with 150 627 new cases for melanoma and over 356 180 for non-melanoma skin cancers. In 2022, the mortality caused by melanoma was 26 360 deaths while for non-melanoma skin cancers was of 12 679 amongst Europe [4]”.
A note has also been added regarding cancer treatment burden to the economy – Introduction, page 1, line 44: “Particularly in the United States (U.S.), the economic burden of this disease was $8.9 billion in recent years (2016-2018). About $6.5 billion were related to non-melanoma tumors, whereas approximately $2.5 billion were associated to melanoma tumors. The latter is expected to triple by 2030 as a result of the overall rising of healthcare costs [10].”.
3. What are the scaleup challenges for the development of nano carriers would be the added advantage to the formulation scientist.
4. Write the recent clinical trials and patents of the same.
Answer: Considering this, notes have been added regarding the scale-up challenges for the development of nano carriers and also on translational challenges – Chapter 4, page 18, line 634: “There are some translational challenges regarding the nanomaterials use in clinics. These challenges can be divided into: i) biological, the internalization and degradation of nanoparticles are important for clinical validation; ii) scale-up, some processes can be less efficient in a large scale (e.g., ultrafiltration); iii) storage, nanomedicines have rigorous storage conditions that are often too expensive for low-income countries to afford; iv) biosafety, when scaling up some changes in the production of nanoparticles can change the quality and safety of the products; v) economic, the costs of production are often high and the relation between nanomaterials costs and consumers purchasing power may lead to product failure in the market [111-113].
At the moment, there is a clinical trial using ASP-1929 combined with anti-PD1 therapy for photoimmunotherapy of recurrent/metastatic head and neck squamous cell carcinoma or for advanced/metastatic cutaneous squamous cell carcinoma. Currently, there are only a few nanomaterials/molecules patented that can be used for skin cancer photoimmunotherapy. The patents for those materials are: US10537641B2, that protects an IR700 molecule conjugated to an antibody that binds HER1; EP3148583B1, that protects an agent comprising a phthalocyanine dye IRDye® 700DX conjugated to a probe that has a molecular weight below 10 kDa; WO2022239766A1, that protects an antibody that binds to CADM1 (Cell Adhesion Molecule 1) on the cell surface; and WO2020163656A1, that protects a formulation that contains a DR6 peptide.”
5. Explain the different modalities (small and large molecules) encapsulation techniques for skin cancer treatment.
Answer: We appreciate the suggestion provided. A paragraph has been added regarding distinct encapsulation strategies used, in Section 2.2, page 6, line 230: “Encapsulation is the method of enclosing molecules inside a shell or carrier, that can be polymers or lipids. It can be used to protect the molecules from degradation, preserve their properties as well as to control their release in the body. Encapsulation can help to deliver the molecules effectively, with a controlled rate, helping to reduce the potential side effects. There are several encapsulation techniques, and the appropriate choice depends on the specific molecule, such as its size, solubility, surface charge, and the therapeutic effect desired. Examples of encapsulation techniques to produce polymer-based nanoparticles are nanoprecipitation or emulsion evaporation; for liposomes production techniques such as hydration of a thin film or solvent injection can be used [87-89]. Nanoprecipitation is a simple technique where non-water-soluble polymer and drug(s) are dissolved in the same organic solvent. The resultant dispersion is added to an aqueous solution that contains a surfactant under stirring, the solvent diffuses into the aqueous phase, causing the polymer to precipitate and form the nanoparticles [88,90]. The emulsion evaporation method consists in dissolving the polymer and drug in a water-immiscible organic solvent that is later added to an aqueous phase containing a suitable stabilizer. Then, the organic solvent is usually evaporated under stirring to obtain the nanoparticles. High-shear mixing can be used to decrease particle size to the nanometric scale and to narrow size distribution [88,89]. The hydration of a thin film technique involves dissolving the lipids in an organic solvent, the solvent is then evaporated to form a thin lipid film. Finally, hydration with water or a buffer is performed to enable the formation of a liposomes dispersion. Sonication is usually performed to increase uniformity and reduce particle size. The solvent injection technique consists of dissolving phospholipids in an organic solvent (e.g., ethanol, ether) that is then injected into an aqueous buffer solution. Lipids self-assembly forming liposomes following ethanol dilution or ether evaporation. This method presents the advantage of allowing to obtain small nanoscale liposomes (usually below 100 nm) with a narrow size distribution, without extrusion or sonication [88].”.
Reviewer 3 Report
This review relates to the use of light for treatment and photosensitizing agents for treatment of skin cancer. Photosensitizing agents are used for two purposes: [1] production of reactive oxygen species with a lethal intent (PDT) or production of heat (PTT). One issue is the ability of light at different wavelengths to penetrate into tissues. This increases with increasing wavelength, with 630 nm being considered as a minimal requirement. Where tumor thickness is confined to only a few cell diameters, green light (500 nm) can be effective. Highly-pigmented tumors, e.g., melanoma, will require longer wavelengths for adequate penetration.
Immunologic effects can also occur where the evoking of an immune response can affect metastatic tumor outside of the light field. This review discusses a collection of delivery systems used to sensitize tissues to light. One requirement is that there be selectivity for neoplastic tissues. Otherwise, the effect can be non-selective.
A few issues detract from the presentation. Data are often presented to 3-4 significant figures, e.g., line 239 and line 363. No biological assay has such accuracy: mean ± standard deviation is more accurate. Some data represent the results of in vitro studies and some relate to animal tumor models or clinical studies. It is not always clear which is which. Mention is often made of ‘viability’ but it is not always clear what test was used: proliferation, clonogenic assays, tumor weight, etc. The Abstract refers to ‘high mortality’ but this applies to only a few forms of cancer, e.g., melanoma. The term ‘phototherapy’ is often used; this is poorly defined. It is preferable to use defined terms: photodynamic therapy and photoimmunotherapy, so as to avoid confusion with phototherapy which is often used to describe the use of blue light for treatment of hyperbilirubinemia.
Author Response
Manuscript ID: biomedicines-2338753
Title: Nanomaterials for Skin Cancer Photoimmunotherapy
Authors: Carlota M. Relvas, Susana G. Santos, Maria J. Oliveira, Fernão D. Magalhães, Artur M. Pinto
We sincerely appreciate all comments and suggestions received and have now carefully replied to all reviewer’s comments point by point and edited the manuscript text accordingly. We believe that the quality and impact of the work has been significantly improved thanks to it and hope that now you find it suitable for publication in Biomedicines Journal. All changes in the manuscript text have been highlighted in green.
Reviewer 3 Comments:
This review relates to the use of light for treatment and photosensitizing agents for treatment of skin cancer. Photosensitizing agents are used for two purposes: [1] production of reactive oxygen species with a lethal intent (PDT) or production of heat (PTT). One issue is the ability of light at different wavelengths to penetrate into tissues. This increases with increasing wavelength, with 630 nm being considered as a minimal requirement. Where tumor thickness is confined to only a few cell diameters, green light (500 nm) can be effective. Highly-pigmented tumors, e.g., melanoma, will require longer wavelengths for adequate penetration.
Immunologic effects can also occur where the evoking of an immune response can affect metastatic tumor outside of the light field. This review discusses a collection of delivery systems used to sensitize tissues to light. One requirement is that there be selectivity for neoplastic tissues. Otherwise, the effect can be non-selective.
1. A few issues detract from the presentation. Data are often presented to 3-4 significant figures, e.g., line 239 and line 363. No biological assay has such accuracy: mean ± standard deviation is more accurate.
Answer: Indeed, this is a pertinent point, and we thank the reviewer for pointing it out. We presented the results as mean ± standard deviation (SD), or mean rounded up to units when SD values were not available. All changes are highlighted in green in the text.
2. Some data represent the results of in vitro studies and some relate to animal tumor models or clinical studies. It is not always clear which is which. Mention is often made of ‘viability’ but it is not always clear what test was used: proliferation, clonogenic assays, tumor weight, etc.
Answer: We appreciate your suggestion and have carefully revised the manuscript accordingly. It has now been clearly indicated along the text when we talk about in vitro or in vivo tests. Moreover, the type of tests used has always been indicated now. All changes are highlighted in green along the text.
3. The Abstract refers to ‘high mortality’ but this applies to only a few forms of cancer, e.g., melanoma.
Answers: The manuscript text has been changed to clarify this question: “The relatively high mortality in melanoma and the existing recurrence rates, both for melanoma and non-melanoma, create the need for studying and developing new approaches for skin cancer management.”
4. The term ‘phototherapy’ is often used; this is poorly defined. It is preferable to use defined terms: photodynamic therapy and photoimmunotherapy, so as to avoid confusion with phototherapy which is often used to describe the use of blue light for treatment of hyperbilirubinemia.
Answer: We appreciate the suggestion, and the term phototherapy has been now replaced with more suitable terms along the text, changes have been highlighted in green.
Round 2
Reviewer 2 Report
The review modified as we the advised edits. No further comments.